# Interface Formation during Collision Welding of Aluminum

**Benedikt Niessen [1],\***, **Eugen Schumacher [2]**, **Jörn Lueg-Althoff [3]**, **Jörg Bellmann [4,5]**, **Marcus Böhme [6]**, **Stefan Böhm [2]**, **A. Erman Tekkaya [3]**, **Eckhard Beyer [4]**, **Christoph Leyens [5,7]**, **Martin Franz-Xaver Wagner [6]** and **Peter Groche [1]**

[1] Institute for Production Engineering and Forming Machines—PtU, The Technical University (TU) of Darmstadt, Otto-Berndt-Strasse 2, 64287 Darmstadt, Germany; groche@ptu.tu-darmstadt.de

[2] Department for Cutting and Joining Manufacturing Processes—tff, The University of Kassel, Kurt-Wolters-Str. 3, 34125 Kassel, Germany; e.schumacher@uni-kassel.de (E.S.); s.boehm@uni-kassel.de (S.B.)

[3] Institute of Forming Technology and Lightweight Components, TU Dortmund University, Baroper Str. 303, 44227 Dortmund, Germany; joern.lueg-althoff@iul.tu-dortmund.de (J.L.-A.); erman.tekkaya@iul.tu-dortmund.de (A.E.T.)

[4] Institute of Manufacturing Science and Engineering, Technische Universität Dresden, George-Baehr-Str. 3c, 01062 Dresden, Germany; joerg.bellmann@tu-dresden.de (J.B.); eckhard.beyer@tu-dresden.de (E.B.)

[5] Fraunhofer IWS Dresden, Winterbergstr. 28, 01277 Dresden, Germany; christoph.leyens@tu-dresden.de

[6] Institute of Materials Science and Engineering, Chemnitz University of Technology, Erfenschlager Straße 73, 09125 Chemnitz, Germany; marcus.boehme@mb.tu-chemnitz.de (M.B.); martin.wagner@mb.tu-chemnitz.de (M.F.-X.W.)

[7] Institute of Materials Science, Technische Universität Dresden, Helmholtzstr. 7, 01069 Dresden, Germany

\* Correspondence: niessen@ptu.tu-darmstadt.de; Tel.: +49-6151-16-23148

**Abstract:** Collision welding is a high-speed joining technology based on the plastic deformation of at least one of the joining partners. During the process, several phenomena like the formation of a so-called jet and a cloud of particles occur and enable bond formation. However, the interaction of these phenomena and how they are influenced by the amount of kinetic energy is still unclear. In this paper, the results of three series of experiments with two different setups to determine the influence of the process parameters on the fundamental phenomena and relevant mechanisms of bond formation are presented. The welding processes are monitored by different methods, like high-speed imaging, photonic Doppler velocimetry and light emission measurements. The weld interfaces are analyzed by ultrasonic investigations, metallographic analyses by optical and scanning electron microscopy, and characterized by tensile shear tests. The results provide detailed information on the influence of the different process parameters on the classical welding window and allow a prediction of the different bond mechanisms. They show that during a single magnetic pulse welding process aluminum both fusion-like and solid-state welding can occur. Furthermore, the findings allow predicting the formation of the weld interface with respect to location and shape as well as its mechanical strength.

**Keywords:** collision welding; impact welding; magnetic pulse welding; model test rig; welding window; jet; cloud of particles; welding mechanisms

## 1. Introduction

One of the biggest challenges today is climate change and its impact on the environment and human society. Driven by politics and self-motivation, the industry is increasingly striving for sustainable

products and manufacturing processes, e.g., by introducing clean production methods without toxic components and with less energy and raw material consumption. Furthermore, new products have to be more environmentally friendly. For example, the consistent lightweight design in the transport sector is an important factor to reduce emissions. Reliable joining techniques are key to implement load-adapted material usage and to fulfill further operational functions. Conventional fusion-based joining processes, however, reach their technological limits when it comes to metallurgical joining between dissimilar metals. In contrast, solid-state welding techniques like magnetic pulse welding (MPW) can lead to advantageous properties like high bond strengths, no heat-affected zones and low electrical resistance, even between metals with differing thermomechanical and chemical properties [1,2]

MPW is based on the oblique collision between two joining partners at high relative velocities [3], thus belonging to the category of collision welding processes like explosion welding or laser impact welding. Usually, one of the joining partners, called flyer, is accelerated up to several hundred meters per second and collides with a stationary so-called target at an impact velocity $v_{\mathrm{imp}}$ under a collision angle $\beta$ (see Figure 1). Due to this angle, a collision front (or in the two-dimensional case a point of collision (PoC)) moves along the colliding surfaces characterized by the collision point velocity $v_{\mathrm{c}}$. High strain rates of up to $10^6$ 1/s and high pressures of up to several GPa occur at the collision point [4,5]. When the dynamic elastic limit of the material is exceeded, material flow results from the plastic deformation of the contact surfaces and a stream of material is pushed ahead of the collision point, see detail in Figure 1 [6–8]. This phenomenon is called jetting and is, besides other criteria, regarded as a necessary condition for bond formation. The jet can remain as a cumulative stream or can disperse in particles. Furthermore, the extensive local strains at the point of the collision lead to the removal of brittle oxide layers and surface contaminations from the surfaces, which are ejected either as a compact stream or as a dispersed cloud of particles (CoP; see Figure 1) [9]. Depending on the collision conditions, the CoP either results of the dispersed material stream, the spalled surface contamination and oxide layers or both phenomena, whereas the cumulative jet can be partly or completely hidden by the CoP [9,10]. This ejection is typically accompanied by a process glare in the form of a bright light or flash emission [11]. Due to the high pressure and temperature at the PoC, the clean surfaces are forced into intimate contact, which ultimately triggers the bonding mechanism. Cui et al. [12] identified three different joining mechanisms when joining aluminum and titanium by MPW, depending on the prevalent kinetic and thermal energies: solid-phase metallurgical bonding by diffusion, liquid-state bonding by melting and solid-liquid coexistence state bonding.

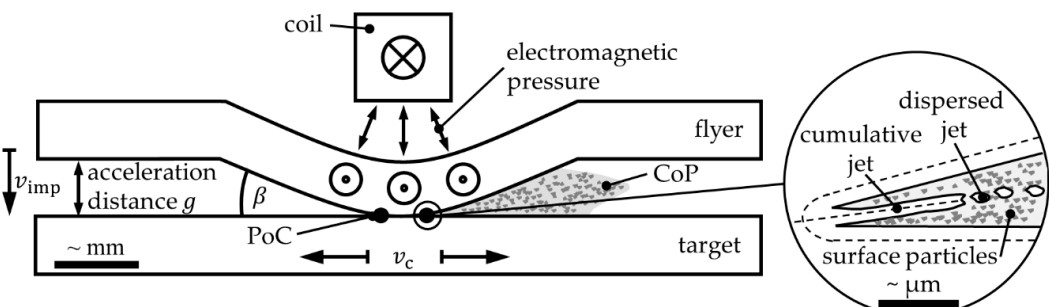

**Figure 1.** Schematic illustration of symmetric sheet magnetic pulse welding (MPW) with the formation of the jet as material flow at the point of collision according to [7,10] in detail. The cloud of particles (CoP) is either formed by dispersed jet particles, spalled contamination and oxide layers or both phenomena.

During collision welding, the process-related acceleration of the joining partners determines the provided kinetic energy. In the case of MPW, the mobile joining partner is accelerated by an induced electromagnetic pressure (see Figure 1) that is generated by closing an electrical circuit that consists of a charged capacitor bank and a coil actuator [3]. The energy input can be adjusted to the welding task by variation of the charging energy. Different coil designs allow welding of overlap joints of profiles, tubes and sheets, respectively. The high repetition rate and the ability to integrate the system into

production lines are two main advantages for the usage of MPW in mass production [13]. However, the joining process is not yet widely applied. One reason for this is the necessary certification of the joining properties, which, in addition to characterization of joint strength, also requires the verification of fatigue strength, corrosion behavior or gas tightness, for instance. Recently, these issues have been objects of concentrated research, and some promising results have already been obtained [14–18].

In addition, a deeper understanding of the joint behavior under different loading scenarios during service is essential for an adequate design of the joining partners and the joint. Ideally, this would cover the prediction of the weld interface position, its shape and area as a function of the material and process parameters. Therefore, a wider comprehension of the bond formation and its influencing parameters is necessary. However, the collision parameters change along the propagating collision front due to the transient behavior during MPW and, thus, cannot be simply calculated or measured [19–21]. In addition, the nature of the process hinders a separate investigation of the influence of the velocity, the mass and the resulting effective energy, which result from the selected acceleration distance and the charging energy of the MPW setup [22,23]. Thus, no direct correlation between the energy input and the welding result can be drawn. Therefore, a purely mechanical model test rig (described in Section 2.1) was designed that allows an independent adjustment of impact velocity and collision angle [24]. A study with copper-copper joints on weld interface formation depending on these parameters showed that after reaching a minimum velocity, the shape and size of the weld interface was influenced particularly by the collision angle [25]. Higher impact velocities increase the region where large weld interfaces can be produced and shift this region towards larger collision angles [25].

The role of the energy input for bond formation is, however, still unclear. This includes additional effects which occur when the kinetic energy is varied either by changing the accelerated mass or the impact velocity. Former investigations mostly varied the kinetic energy of the flyer via the impact velocity at a constant flyer mass. However, this also changes the collision conditions. For explosion welding, Lysak et al. [26] related the collision conditions to the energy part for the metal's plastic deformation by adding the averaged mass to the classical welding window as a third dimension. Thereby the hydrodynamic processes of the collision are linked to the metallophysical processes of bond formation.

To transfer these findings to the MPW process at comparatively low energy input, the influence of the energy input is examined in this paper separately from the collision conditions by changing the flyer mass and keeping the impact velocity constant. For this purpose, several experimental series with flyers of three different thicknesses are carried out using two different experimental setups. The experiments on the mechanical model test rig serve to change the particular parameters individually and to determine their influence on the weld interface, they are supplemented by experiments with different impact velocities. For this purpose, the different phenomena occurring at collision welding can be related to the formation of the weld interface for different points in the welding window. Subsequently, the results are validated by experiments on an MPW setup with different flyer thicknesses, for which the necessary impact velocity parameter settings are determined with an adapted measurement setup. This allows the determination of the influence of different energy inputs at otherwise comparable collision conditions. Based on the experimental results, the following questions are addressed in the present paper:

(1) How does the kinetic impact energy of the flyer influence the collision process and the formation of the weld interface and how is the welding window affected?

(2) Do the collision kinetics influence the governing phenomena, the resulting bond mechanism and thus the weld interface's properties, e.g., mechanical strength?

(3) Can the formation and properties of the weld interface be predicted with respect to location, shape and strength, and how can these properties be controlled by the process parameters?

## 2. Materials and Methods

### 2.1. Description of the Series of Experiments

Three series of experiments, performed in two different setups, a model test rig (see Section 2.2) and MPW setup (see Section 2.3), were carried out with aluminum sheets (EN AW-1050A Hx4, yield strength: 99 MPa, tensile strength: 105 MPa) with an initial thickness of $s = 2$ mm for the target as well as the flyer. The specimens for the test rig were produced by laser cutting, while specimens for the MPW setup were cut to size (40 mm × 100 mm) by plate shears. To study the influence of the flyer thickness while keeping the material properties constant, the sheet thickness for both setups was reduced by milling to 1 and 1.5 mm, respectively. The main experiments were carried out with an impact velocity of 262 m/s (see Table 1, Series 1.1 and 2.1., 2.2, 2.3). Furthermore, a second series of complementary experiments were carried out in the test rig with a target and flyer thickness of 2 mm and varied impact velocities of 220 m/s and 240 m/s, respectively (see Table 1, Series 1.2).

**Table 1.** Summary of series of experiments in applied setups with varied and constant parameters.

| Series of Experiment | Applied Setup | Varied Key Parameter | Constant Parameter |
|---|---|---|---|
| 1.1 | Test rig | Flyer thickness ($s = 1$ mm, 1.5 mm, 2 mm) Collision angle ($\beta = 3$–9°) | $v_{imp} = 262$ m/s |
| 1.2 | Test rig | Impact velocity ($v_{imp} = 220$ m/s, 240 m/s) Collision= 3–9°) | $S = 2$ mm |
| 2.1 | MPW | Flyer thickness ($s = 1$ mm, 1.5 mm, 2 mm) | $v_{imp} = 262$ m/s, g = 1.5 mm |
| 2.2 | MPW | Flyer thickness ($s = 1$ mm, 1.5 mm, 2 mm) | $v_{imp} = 262$ m/s, g = 2.0 mm |
| 2.3 | MPW | Flyer thickness ($s = 1$ mm, 1.5 mm, 2 mm) | $v_{imp} = 262$ m/s, g = 2.5 mm |

In the test rig, the collision angle was varied to define the weldable region of the welding window. From earlier investigations, it was known that collision angles that led to welding are in the range of 3° to 9° for 2 mm thick joining partners of aluminum at an impact velocity of 262 m/s. Based on this, the weldable range was determined for each flyer thickness and impact velocity. The joints are considered welded if they cannot be separated manually after the experiments.

The initial impact velocity was adjusted close to 262 m/s using the MPW setup by means of photonic Doppler velocimetry (see Section 2.4.3 and Table 1). As mentioned above, the further progression of impact velocity and collision angle was unsteady and difficult to measure in this setup. However, the initial collision angle and its rate of change along the propagating collision front was varied by different acceleration distances of 1.5 mm, 2 mm and 2.5 mm.

### 2.2. Model Test Rig

Besides the individual and precise adjustment of the collision parameters at stationary process conditions, the model test rig was built up (by the PtU Institute, Darmstadt, Germany) with the intention to provide good observability, which was realized by a purely mechanical concept (see Figure 2a). The main components are two rotors with a diameter of 500 mm, each one driven by a synchronous motor. As a joining partner, specimens with a collision area of 12 mm × 12.5 mm were mounted and prebent with a certain angle at one end of each rotor (see Figure 2b,c). In order to start the collision welding operation, both rotors rotated in the same turning direction but with a phase offset of 45°. As the rotational speed reached half of the desired impact velocity, the phase offset was compensated within one revolution. Thus, the two specimens collided with high accuracy and repeatability in the center between the two turning points. After the collision and the accompanying welding process, the specimens were torn off at the predetermined breaking point since the rotors could not be stopped instantaneously; see Figure 2d. The applied configuration of the test rig for these experiments led to a maximum absolute impact velocity $v_{imp}$ of 262 m/s [27].

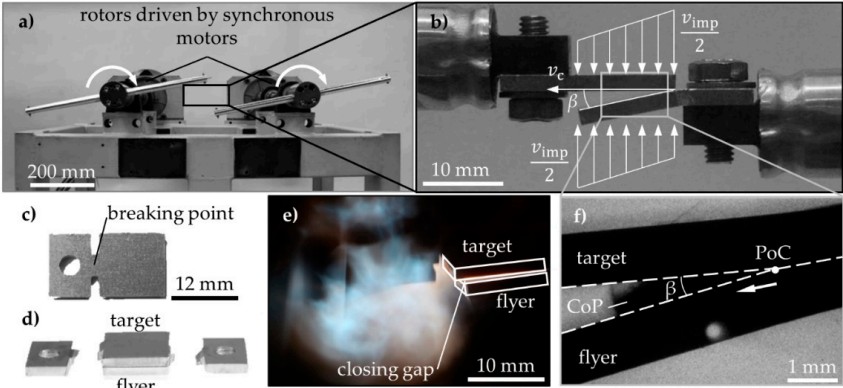

**Figure 2.** (**a**) Test rig assembly [28], (**b**) shown in detail: mounted joining partners and the resulting process parameters at the moment of the initial impact [28]. (**c**) Geometry of the specimen in the test rig [24] and (**d**) welded specimens [28]. (**e**) A long-term exposure of process glare (2 s exposure time) and (**f**) high-speed image [24] of the collision welding process with 20 ns exposure time were recorded. (a,b,d) are reproduced from [28], with permission from Elsevier, 2017; (c,f) are reproduced from [24], with permission from John Wiley and Sons, 2019. The entire figure is also published in the companion paper [10].

## 2.3. MPW Setup

The MPW experiments were carried out with the pulse generator BlueWave PS48–16 from PSTproducts GmbH, Alzenau, Germany in combination with the sheet welding tool coil B80/10. The pulse generator provides a maximum charging energy of 48 kJ and a maximum charging voltage of 16 kV. The effective part of the tool coil had a width of 10 mm, a length of 80 mm, and a thickness of 5 mm. It can be operated up to a maximum peak current of 500 kA. During the welding experiments, the flyer and target sheets were positioned in the center above the coil with an overlap of 30 mm and were fixed by a steel backing plate (see Figure 3).

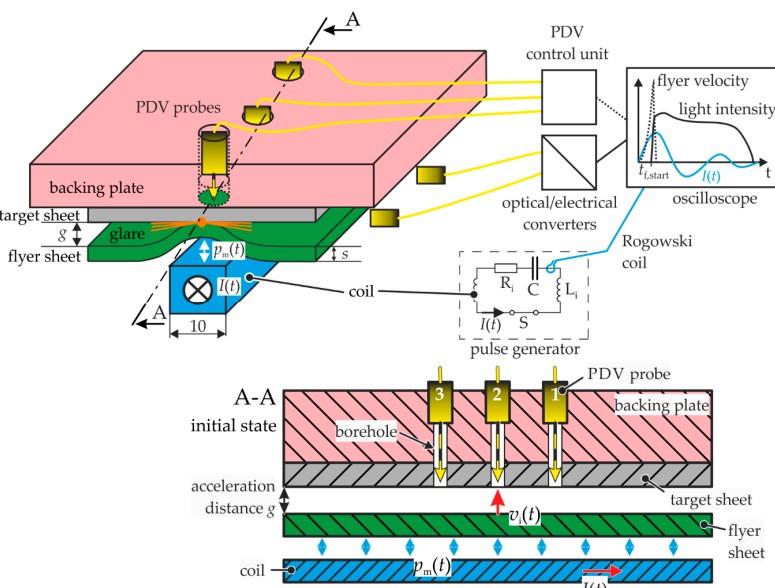

**Figure 3.** MPW setup for sheet welding: the acceleration gap *g* and the flyer sheet thickness *s* were varied. Section A-A shows the photonic Doppler velocimetry (PDV) measurement configuration at the initial state of the welding process (no deformation of the flyer). The oscilloscope recorded the signals of the PDV system, Rogowski coil and measured intensities of the process glare.

## 2.4. Methods of Process Observation

### 2.4.1. Process Observation in the Model Test Rig

Two observation methods were implemented in the test rig. First, the collision welding process was observed by an image intensifier camera hsfc pro (by PCO, Kelheim, Germany) with a long-distance microscope lens. This system allowed taking up to eight images per experiment. Due to the high-velocity collision an exposure time of 20 ns was applied (Figure 2f). During the experiments, a CAVILUX Smart lighting laser (by Cavitar, Tampere, Finland) with a power of 400 W and a wavelength of 640 nm provided sufficient brightness. In combination with an optical bandpass filter placed in front of the camera lens, the bright process glare was suppressed, which would otherwise outshine the phenomena in the closing gap. A script in MATLAB (by MathWorks, Natick, Massachusetts, MA, USA) was used to measure the collision angle $\beta$ by edge detection in each high-speed image, as shown exemplarily in Figure 2f [24,28,29].

Second, a qualitative examination of the process glare was realized by long-term exposures (Figure 2e), with a single-lens reflex camera 5D (by Canon, Ōta, Tokio, Japan) and a 100 mm macro lens, which was positioned at an angle of about 45° to the rotor center axis. The image acquisition settings were set to an exposure time of 2 s, an aperture of F13 and a light sensitivity of ISO 100. The results are shown in Appendix A.

### 2.4.2. Rogowski Coil

The recording of the discharge current was performed via a Rogowski pickup coil type *CWR* 3000 B (by Power Electronic Measurements Ltd., Long Eaton, Nottingham, UK) in combination with a high-resolution oscilloscope (see Figure 3). Rogowski coils are especially suitable for the measurement of oscillating currents with high amplitudes and frequencies like those occurring during MPW. The discharge current curve $I(t)$ contains important information for the evaluation of the temporal evolution of the MPW process and is shown in Figure 4, together with the recorded light intensity.

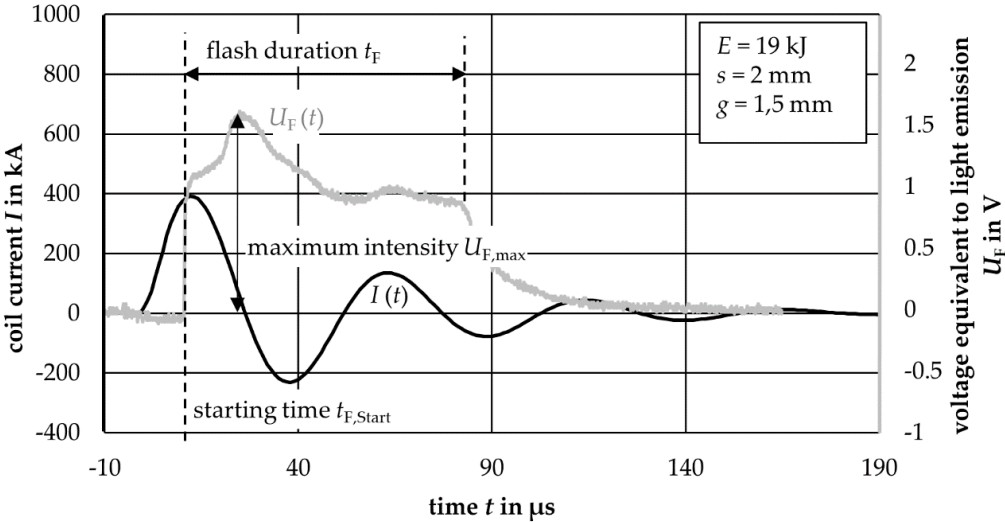

**Figure 4.** Signals of the Rogowski pickup coil and the flash measurement system showing the time-dependent evolution of the tool coil current and of the light emission, respectively, in the setup with the pulse generator BlueWave PS48-16.

### 2.4.3. Photonic Doppler Velocimetry

An accurate, quantitative determination of the flyer velocity at the moment of impact is crucial for the setting of similar impact velocities at the different experimental setups. Compared to the test rig, the accessibility of the collision zone during the MPW process is very limited for cameras due to the

small acceleration distance and the progression of the collision front. The initial impact velocity at a fixed acceleration distance can be adjusted via the charging energy, but the analytical or numerical determination is elaborate and the MPW process is very sensitive to small disturbances like variations of material properties or flyer dimensions. Therefore, photonic Doppler velocimetry was applied for the measurement of the impact velocity $v_{imp}$ during the MPW experiments. This robust and accurate method, developed by Strand et al [30], is based on the laser Doppler effect. It is an established measurement technology in the field of high-velocity forming and joining [31]. With the applied PDV system, velocities of up to 1.1 km/s can be measured using three parallel channels. The characteristics of the applied system are described by Lueg-Althoff in [32].

　　The application of PDV measurements requires the direct accessibility of the surface of the moving object, i.e., the flyer part in MPW. Therefore, three focuser probes were positioned normal to the setup (see Figure 3). Small boreholes were drilled into dummy target parts and the backing plate in order to allow the laser beams to directly illuminate the moving flyer surface. This allows measuring the temporal evolution of the flyer velocity from the beginning of the movement until the impact, and to adjust the impact velocity $v_{imp}$ for all three flyer thicknesses and acceleration distances by modifying the charging energy of the pulse generator and the corresponding current flow in the tool coil, respectively. In Table 2, the determined charging energies are listed for several combinations of flyer thickness and acceleration gap (Experimental Series 2.1, 2.2, 2.3), as well as the averaged measurements of maximum current, discharge frequency and impact velocity (including minimum and maximum values). The boreholes in the target plate inhibited welding between the flyer and the dummy target parts. Nevertheless, the evolution of the flyer impact velocity was not affected by the presence of the boreholes until the collision because the magnetic field is completely shielded by the conductive flyer part. Therefore, it was assumed that the normal impact velocity $v_{imp}$ determined at the three measuring points was identical in the experiments with a solid target for the real welding experiments.

**Table 2.** Determined charging energies for combinations of flyer thickness and acceleration gap and averaged measurements of maximum current, discharge frequency and impact velocity.

| Flyer Thickness $s$ in mm | Acceleration Gap $g$ in mm | Charging Energy in kJ | Ø max. Current in kA | Ø Discharge Frequency in kHz | Min. Impact Velocity in m/s | Max. Impact Velocity in m/s | Ø Impact Velocity in m/s |
|---|---|---|---|---|---|---|---|
| 2.0 | 1.5 | 19.0 | 392.4 | 19.7 | 255 | 261 | 257.0 |
| 2.0 | 2.0 | 18.0 | 381.3 | 19.7 | 255 | 258 | 256.3 |
| 2.0 | 2.5 | 17.5 | 376.0 | 19.7 | 259 | 262 | 260.7 |
| 1.5 | 1.5 | 14.5 | 340.5 | 19.7 | 251 | 257 | 254.3 |
| 1.5 | 2.0 | 13.7 | 330.4 | 19.7 | 251 | 256 | 253.0 |
| 1.5 | 2.5 | 13.3 | 325.3 | 19.7 | 249 | 255 | 252.3 |
| 1.0 | 1.5 | 9.7 | 274.6 | 19.7 | 261 | 269 | 263.7 |
| 1.0 | 2.0 | 9.3 | 268.7 | 19.7 | 260 | 261 | 260.3 |
| 1.0 | 2.5 | 8.7 | 259.8 | 19.7 | 249 | 250 | 249.3 |

### 2.4.4. Flash Detection

　　High-speed collision processes are accompanied by a characteristic flash, which is called impact flash [33]. During the MPW experiments, the time-dependent evolution of the light emission was measured with the flash measurement system explained in [34]. Figure 4 shows an example of the time-resolved light intensity as well as the derived characteristic values—the starting time of the flash, its duration and maximum intensity. According to previous studies, the flash duration was defined as the duration between the initial increase of the light intensity and its steep decrease [34]. These parameters were taken with two independent sensors at the same distance to the welding zone of approximately 15 mm and were then averaged for each series of experiments with a specific flyer thickness and acceleration gap, respectively. The results are shown in Appendix A.

### 2.5. Analysis of the Weld Interface

　　A nondestructive analysis method of the welded area was carried out using a 2D ultrasonic measurement with the MiniScanner (by Amsterdam Technology, Zwinderen, The Netherlands) for experiments on the model test rig. This device scans an area of 12 mm × 25 mm during a single scan

run in pulse-echo mode with a local resolution of approximately 0.1 mm × 0.1 mm spots. For each spot, a so-called A-scan echo was recorded with its two-dimensional coordinates. This scan information was analyzed using a MathWorks MATLAB script. Depending on the signal, a differentiation was made between "bond", "no bond" or "no information" for each scanned point. This approach delivered both a qualitative and quantitative result in terms of the welded area [24]. Since the joining partners tear off after their collision in the model test rig, subsequent collisions with the still rotating rotors and the housing can occur. These joints were post-treated by flattening prior to ultrasonic examination. In addition, scratches and burrs were removed from the surfaces by grinding to improve the signal quality. If the deformation exceeds a certain limit, the evaluation of the joint was affected and hindered locally or completely.

The joints produced at the MPW setup were tested for their bond strength by tensile shear testing in a Z100 testing machine (by Zwick, Ulm, Germany) with three repetitions for each parameter set at a testing velocity of 10 mm/min. Additionally, for one joint of each series, a cross-section was prepared parallel to the central plane in the welding direction as shown in Figure 5. Due to the symmetric collision of the flyer with the target, there were two symmetric points of collision that moved in opposite directions and formed two weld interfaces; see also Figure 1. To characterize the welding result, the widths of the two welding interfaces and the gap between them were measured in the cross-section using an optical microscope (OM) DM2700 (by Leica Microsystems, Wetzlar, Germany).

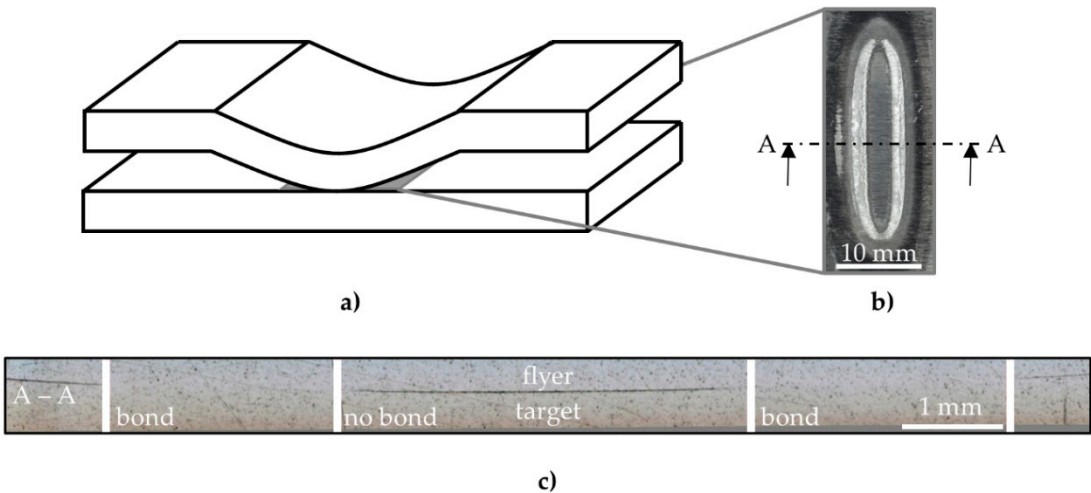

**Figure 5.** (**a**) Schematic sketch of a welded MPW joint (**b**) with a fracture image of the welding interface with typical elliptical ring-shape after tensile shear test. The location of the cross-section is marked by the dot-dashed line (A–A) and was analyzed by OM and SEM. An exemplary OM-image of a cross-section (A–A) is shown in (**c**).

The microstructures of the weld interfaces of joints made with both setups were further analyzed with a Ultra Plus (by Zeiss, Jena, Germany) scanning electron microscope (SEM), using the secondary electron detector to validate the results of the ultrasonic examination and also to determine how the different weld interface types have been formed.

## 3. Results

### 3.1. Model Test Rig

Figure 6 shows the welding windows for the series of Experiments 1.1 and 1.2 performed in the test rig. Although in both cases, the lower boundary collision angle, below which bond formation is inhibited, did not differ strongly, the upper boundary collision angle increased both with higher flyer thickness and increased impact velocity.

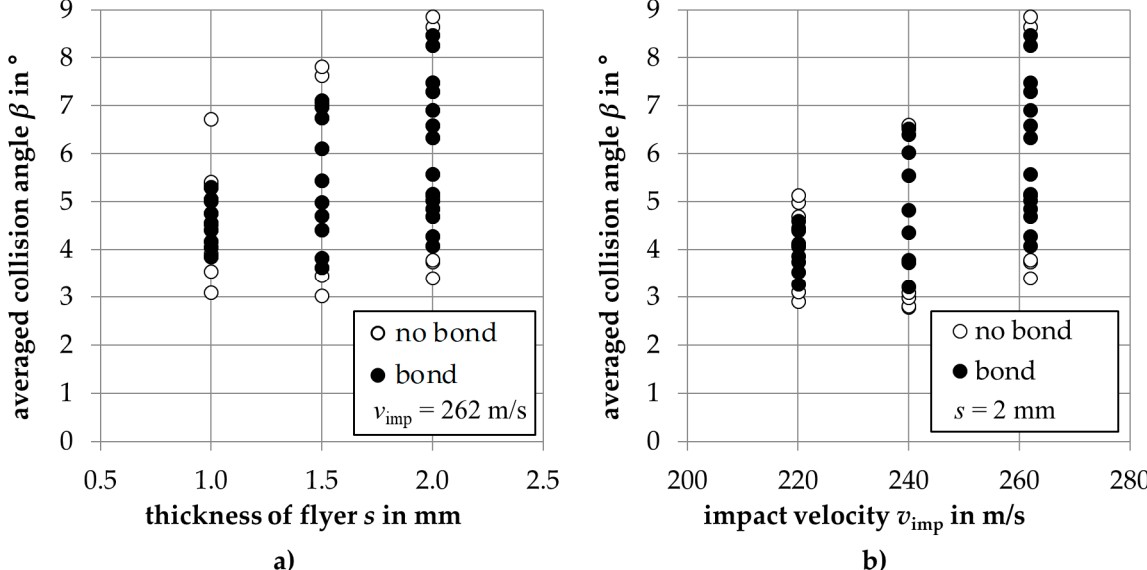

**Figure 6.** Welding windows (**a**) for different flyer thicknesses and collision angles and (**b**) for different impact velocities and collision angles, each point represents one experiment. The upper boundary angle increased in both cases, while the lower boundary varied only slightly.

For Series 1.2, the findings correspond to those already obtained in welding window investigations for another batch of the same material. In this context, the lower boundary angle was related to the suppressed ejection of the CoP, which inhibits bond formation by the reinclusion of the CoP particles at the PoC. In contrast, the upper boundary angle defines the process parameters up to which jet formation can be initiated and sustained [23].

The results of the ultrasonic analysis of the weld interface of Series 1.1 are shown in Figure 7. A similar curve of the ratio between the welded and overlapping areas over the collision angle was achieved for all three flyer thicknesses (see Figure 7a). Just a small amount of the overlapping area was welded close to the lower and upper boundary angles. In between, there is a region where large area welds can be formed. However, none of the specimens was completely welded. The range of this region, its maximum value and the corresponding collision angle increased with increasing flyer thickness and was shifted to larger collision angles. This behavior had also been observed in experiments where copper was welded at different impact velocities [24] and was further validated by the results of Experimental Series 1.2 with lower impact velocities using a flyer thickness of 2.0 mm. The types of weld interfaces described in [25] can also be found here and support the described phenomena at the upper and lower boundary angle. For small collision angles and without inhibiting the bond formation, the CoP could only escape sufficiently in the lateral regions and at the end of the closing gap. At large collision angles, the jet formation was initiated after an entry region, but then broke down while the flyer continued to deform on top of the target (see Figure 7b). However, this weld interface type was not as pronounced regarding the termination of the weld interface formation as in the previous experiments with copper. Furthermore, no completely welded interface could be obtained, which might be due to the fact that the investigations were carried out close to the lower limit of the welding process with respect to the energy input.

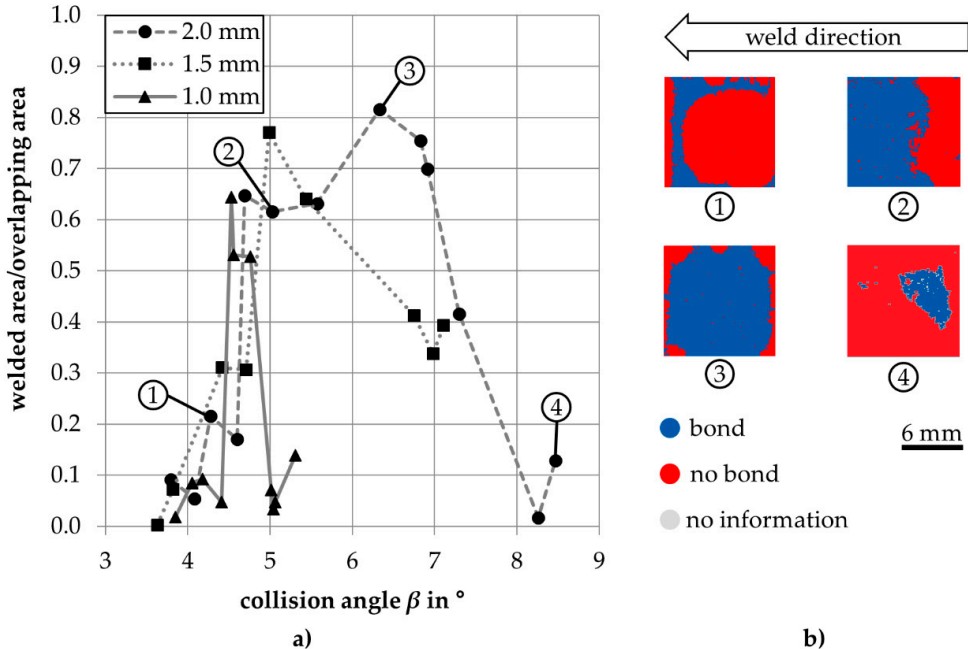

**Figure 7.** (**a**) Progression of welded to the overlapping area over different collision angles for the three flyer thicknesses (target thickness: 2 mm, $v_{\text{imp}}$ = 262 m/s); (**b**) two-dimensional representation of the weld interface obtained from the ultrasonic analysis.

Figure 8 summarizes the SEM analysis of the interfaces of Experimental Series 1.1. The welded interfaces are mostly straight and only single instances of wavy patterns can be found. Furthermore, the findings of the nonwelded interfaces and the transition regions support the hypothesis regarding the boundaries by collision angle and the related mechanisms. In Figure 8a the collision at an angle close to the lower boundary angle for 2 mm flyer thickness started without visible interaction of the surfaces (1). Shortly afterwards, the surfaces were contaminated by the enclosed CoP (2) whose amount increased along the joining gap (3). At a certain stage, the conditions in the gap changed in a way that local melting and resolidification occurred at the surfaces and the surfaces got continuously closer, until the formation of the weld interface began (4, 5). At the end of the weld interface a continuous melted and resolidified interlayer was found (6).

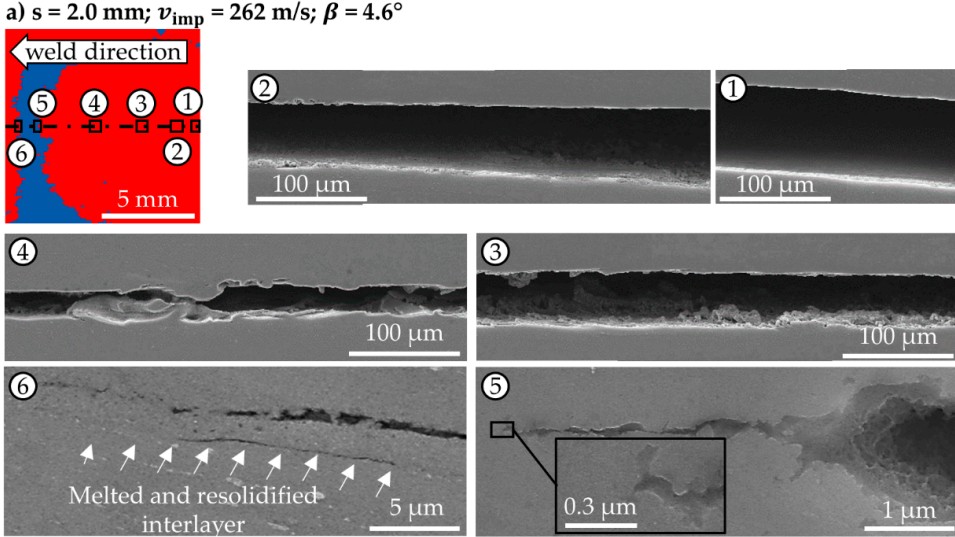

**Figure 8.** *Cont.*

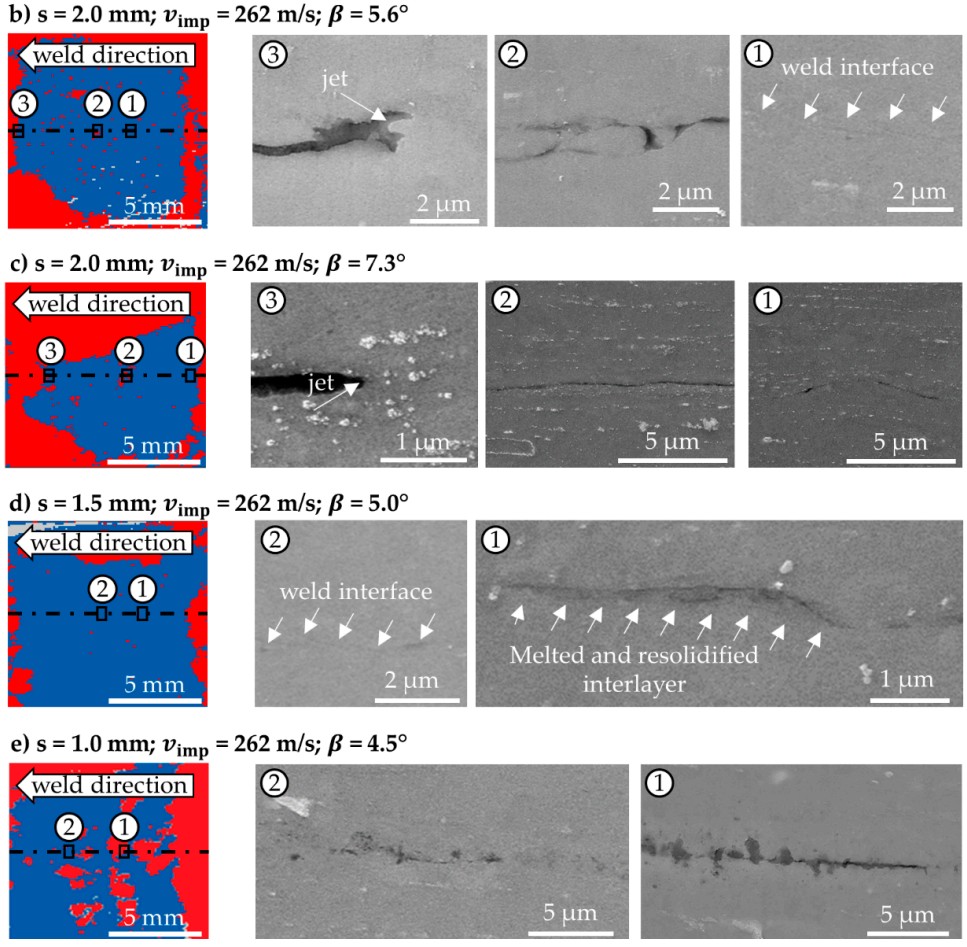

**Figure 8.** SEM analysis of test rig joints at different locations of the weld interface. Blue coloring indicates bond, red indicates no bond. (**a–c**): *s* = 2 mm: (**a**) Close to the lower boundary collision angle (4.6°); (**b**) in the region with a large welded interface (5.6°), (**c**) close to the upper boundary collision angle (7.3°); (**d**) largest welded area (5.0°) for *s* = 1.5 mm; (**e**) largest welded area (4.5°) for *s* = 1 mm (note that different magnifications are used in the SEM micrographs to highlight relevant features).

In the region with a large welded area for 2 mm flyer thickness, the weld interface could hardly be recognized in the SEM micrograph (see Figure 8b) (1); however, nonwelded areas could be clearly identified. Only at the beginning of the weld interface, some melted structures and a porous interface were observed. Later on, the few nonwelded regions in the center did not contain porous material from the CoP but obviously parts of the jet stream which were torn off and rolled over by the PoC and hindered the bond formation (2). Moreover, at the end of the weld interface, the spilled jet was clearly visible (3—arrow).

In joints with 2 mm flyer thickness, produced close to the upper boundary angle, the welded regions were not properly formed and contained several imperfections (Figure 8c) (1, 2). The jet at the end of the weld interface was significantly thinner (3) than in the region described above.

Looking at the joints with 1.5 mm and 1 mm flyer thickness, large welded areas were found. While for *s* = 1.5 mm the sound weld interface was mostly hardly visible in the SEM micrographs (Figure 8d) (2), other parts contained locally melted and resolidified interlayers (1). The weld interface for s = 1.0 mm exhibited partially porous regions, melting structures and cracks (see Figure 8e), which were partly declared as sound weld by the ultrasonic analysis (2). Nonwelded regions showed larger melting defects (1).

### 3.2. MPW Setup

Figure 9 shows the results of the weld interface formation in the MPW setup for the different flyer thicknesses and acceleration distances at selected positions of the weld interfaces (compare Table 1, Series 2.1). Considering the parameters separately, the start of the weld interface was not influenced by the flyer thickness. The end position of the weld interface increased with increasing flyer thickness. When the acceleration distance was enlarged, weld interface formation started earlier, but also ended earlier.

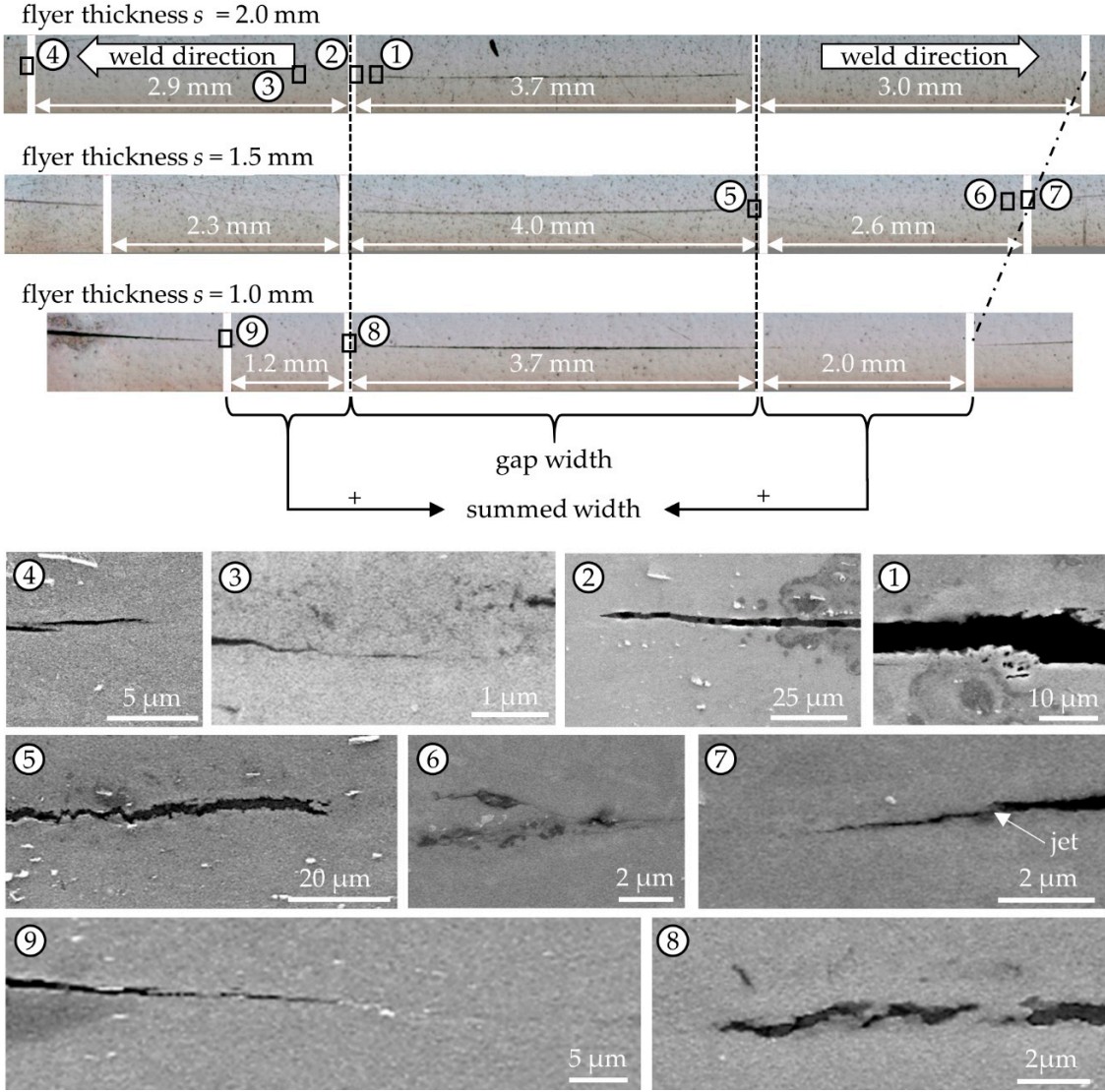

**Figure 9.** Microsections along the central plane parallel to the welding direction of weld interfaces in the MPW setup for an acceleration gap of $g = 1.5$ mm (Series 2.1): While weld interface began at the same position (dashed line), it ends later with increasing flyer thickness (dot-dashed line). Below, the results of the SEM investigation are shown in detail (1–9).

The summed width of both weld interfaces tended to decrease with increasing acceleration distance, especially for smaller flyer thicknesses, see also Figure 10. Furthermore, the flyer thickness of 1 mm exhibited an asymmetric image of the weld interfaces for 1.5 mm and 2.5 mm acceleration distances (Series 2.2, 2.3). The latter was only welded on one side in the sectioned joint (see cross-section in Figure 10), which is visible in the diagram by the smaller total weld interface width and thus,

no gap width. This was either a result of the comparably low energy input or of an asymmetric rolling movement of the flyer due to the clamping situation in the weld setup.

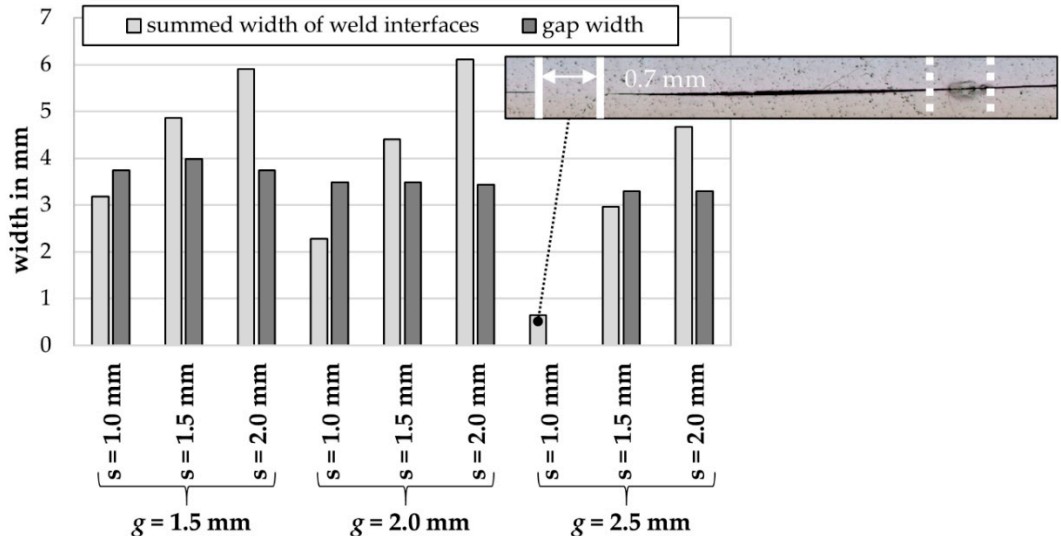

**Figure 10.** Summed width of both magnetic pulse welded interfaces and width of the gap in between for different flyer thicknesses and acceleration gaps at constant impact velocity of 262 m/s. All configurations show the same trend: while the gap width varied slightly for the acceleration distances *g*, the summed width of the weld interface increased with increased flyer thickness *s*. Due to the asymmetric weld formation (see cross-section), no gap width could be determined for the configuration *s* = 1.0 mm and *g* = 2.5 mm.

The SEM analysis of the weld interface in Figure 9 revealed similarities with the results that were produced in the test rig, see Section 3.1. Pores and partly melted and resolidified structures were found in front of (1) and at the beginning of the weld interfaces (2, 5, 8). Such weld defects were related to the heating and/or entrapment of the CoP and indicated a collision angle close to the lower boundary. Similar defects were also located in further sections along the weld interfaces (6). The ends of the weld interfaces (4, 7, 9) were similar to the ones produced in the test rig that were welded at a collision angle close to the upper boundary. A thin jet at the end of the weld was identified for *s* = 1.5 mm (7).

The bearable tensile forces of the different weld configurations (Series 2.1, 2.2, 2.3) are shown in Figure 11. The comparison of the tensile forces revealed that for an acceleration distance of 1.5 mm all joints achieved the bearable tensile force calculated from the tensile strength of the base material and thus, all failed in the base material except for two joints. These two parts showed a nonuniform weld interface formation in the fracture pattern. For joints with a flyer thickness of 2 mm and an acceleration distance of 2 mm, failure occurred in the base material, which was also apparent in the achieved tensile force. In this case, the fracture occurred in the neck region of the flyer close to the welded area, where the cross-sectional area was reduced due to plastic deformation during the welding process.

All other joints failed in the weld interface. Relating the tensile force to the total weld interface width resulted in a ratio of approximately 1.5 kN/mm for all configurations. Only the configurations of 2 mm and 2.5 mm acceleration distance with 1 mm flyer thickness varied to lower values due to the incomplete global weld interface formation. The fracture images of these samples revealed that the weld interface was characterized as two parallel lines instead of a complete elliptical ring (see fracture images in Figure 11). Furthermore, all fracture surfaces showed a symmetric weld interface in contrast to the cross-section in Figure 10. Therefore, the width value of this configuration was corrected by the multiplication by a factor of two to calculate the ratio of tensile force to the width in Figure 11 to represent a symmetric weld interface.

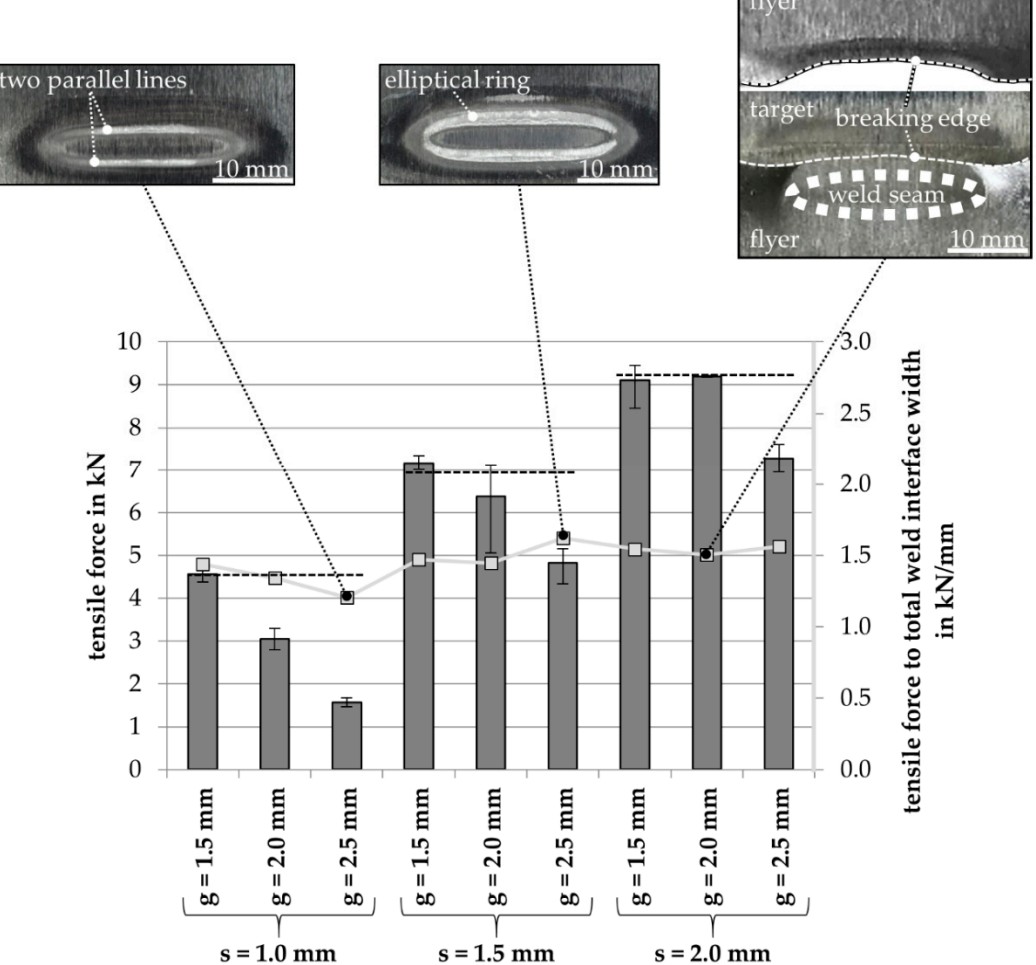

**Figure 11.** Averaged tensile force with minimal and maximal deviation for different flyer thicknesses *s* and acceleration distances *g* (left axis, vertical columns). The dashed lines represent the theoretical bearable tensile force value for the particular flyer thickness calculated by the tensile strength of the base material. The ratio of the tensile force to total weld interface width is represented by rectangles and grey line and the right axis scale. The images of the different fracture surface types in top view are shown to explain the variation of the ratio tensile force to total weld interface width.

## 4. Discussion

### 4.1. Influence of Kinetic Energy Input on Weld Interface and Welding Window

For the comparison of the results of the model test rig and the MPW setup, it is important to recall that the collision conditions continuously change during the MPW process [18–20]. Hence, the impact starts at a small collision angle close to 0° and at the maximum impact velocity. This results in a high collision point velocity. Subsequently, the collision angle increases and the impact velocity decreases until the weldable region of the welding window is entered and then, after further progression of the collision conditions, left again. Depending on the process parameters, the path through the material-specific welding window differs. In the test rig, in contrast, the collision angle and the impact velocity stay constant during the collision. Therefore, each experiment represents a single point in the welding window. Performing a series of experiments with different collision conditions allows defining welding windows for different process and material parameters. This in turn provides an extended understanding of the governing welding mechanisms and delivers additional information for the design of the MPW process with industrial relevance.

Considering the determined welding windows for different thicknesses in the test rig (Series 1.2), it was possible to analyze the formation of the weld interface in the MPW setup (Series 2.1, 2.2, 2.3); compare Figures 6 and 9. The progression of the collision angle was similar for different thicknesses, until the start of the bond formation. In addition, higher thicknesses resulted in longer weld interfaces. For MPW it was not possible to identify, whether the bond formation stopped (i.e. the collision kinetics left the weldable region of the welding window) due to an increase of the collision angle or due to a combined change of collision angle and impact velocity. The latter depends on the different thicknesses, the resulting different stiffness values and thus a varied forming behavior of the flyer during the rolling movement on the target.

When the acceleration distance was increased, the location of the weld interface was shifted closer to the initial point of collision. This was due to the higher progression rate of the collision angle leading to a flyer rolling movement that the lower boundary angle was exceeded earlier and the upper boundary angle was reached faster. This is in good agreement with the findings of Sarvari et al. [21] who investigated different acceleration distances in an MPW setup.

To understand the influence of the mass induced kinetic energy on the bond formation, the results of the test rig experiments with varied thickness (Series 1.1) and varied impact velocity (Series 1.2) are plotted together in the classical $\beta$-$v_c$-welding window in Figure 12. It can be observed that the increase in energy input by changing both the flyer mass and a higher impact velocity led to an expansion of the weldable region towards lower collision point velocities. The change of the impact velocity to achieve a certain kinetic energy also changed the collision conditions while they remained equal when the flyer mass was increased. Moreover, the weldable area of the welding window was affected by the increase in energy. Similar findings were reported by Lysak and Kuzmin in [26] for explosion welding. To explain the solid-state welding mechanism, they related the impact velocity, the collision point velocity and the involved mass to three physical parameters, the pressure at the PoC, the duration of the applied pressure and the temperature in the weld interface zone. In the next section, it is explained how these parameters influenced the phenomena in the joining gap and the governing bond mechanisms in the case of changed energy input.

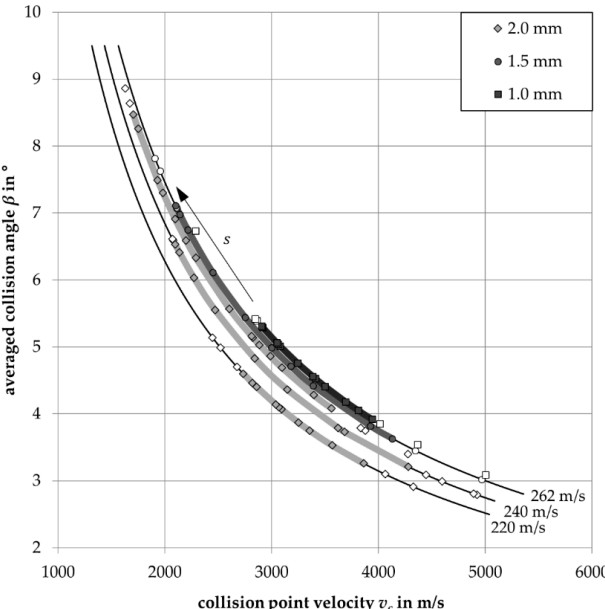

**Figure 12.** Welding window by collision angle $\beta$ over collision point velocity $v_c$ for different flyer thicknesses $s$ and impact velocities $v_{imp}$ measured in the test rig; symbols in shaded graphs represent experiments with bond, symbols outside of the graphs represent experiments with no bond. The different flyer thicknesses at $v_{imp} = 262$ m/s are plotted at slightly different impact velocities to improve the visibility. Arrow indicates the influence of the increasing value on the upper left boundary.

### 4.2. Influence of the Collision Kinetics on the Phenomena and the Bond Mechanisms

The findings of the test rig experiments (Series 1.1 and 1.2) and the analysis of the weld interface revealed that the acting phenomena, especially the role of the CoP, at different points of the welding window have a pronounced influence on the occurring bond formation mechanisms. This has not been taken into account previously. Hence, due to the transient propagation through the welding window during MPW, it can be assumed that different mechanisms of bond formation apply in certain sections depending on the process parameters and the formed CoP.

During test rig experiments with small collision angles, maximum flyer mass and high impact velocity, the CoP had a strong impact on the bond formation. The SEM images in Figure 8a (1–3) for $s = 2$ mm indicate that at first, due to the small angle and resulting high flow resistance, the CoP could not be ejected out of the gap, was entrapped and hindered the contact of the activated base materials. It was found by the analysis of the associated process glare that the CoP can reach temperatures of several thousand Kelvin (see the accompanying paper [10] and compare it with Figure A1). This might also result in the nonwelded regions due to excessive melting of the surfaces (4). Later on, during the collision front propagation towards the free end of the flyer, the resistance to eject the CoP was low enough with the result that the interfaces came into contact and formed a bond (5). In this case, the mechanism of bond formation was attributed to fusion-like bonding, regarding the estimated high-temperature development and no visible signs of a jet in terms of a metal stream. This hypothesis is in good agreement with the findings of Bellmann et al. [35], who observed no deformation of the surface-near layers in the form of a jet at small collision angles but melting close to and in the weld interface.

In contrast, the formation of a jet occurred at larger collision angles (Figure 8b). In the case of large welded areas (ratio: 0.82 to the total overlapping area) in the middle of the estimated welding window for $s = 2$ mm (Figure 8b), most parts of the weld interface exhibited only a few defects (2) and were almost not distinguishable from the base material in the SEM. A determination of the governing bond mechanism was thus not possible. The absence of porous structures at the weld interface may indicate solid-state welding. At this point, it should be noted that the present investigation focused on joints between similar metals. No additional aspects were considered that would occur during the welding of dissimilar metals, like the formation of intermetallic phases.

The weld interface close to the upper boundary angle for $s = 2$ mm exhibited a less distinct jet and several areas without bond (see Figure 8c). This indicates that the deformation of the surfaces was considerably smaller and less adjacent base material was deformed compared to smaller collision angles. Two approaches provide possible explanations for these findings: First, according to Manikandan et al. [36], the depth of deformation of the adjacent base material by the colliding surfaces depends on the induced kinetic energy. Therefore, it can be argued that for large collision angles less induced energy was available for the microscopic deformation close to the point of collision. This can be explained by the fact that at larger collision angles more work was needed to close the joining gap by the continuous bending of the flyer. A second explanation may be the different interactions of the CoP with the surfaces in front of the point of collision. As mentioned above, the CoP can reach very high temperatures. Even if it did not melt the surfaces, the induced heat could cause a considerable reduction of the flow stress close to the point of collision, allowing more material to flow in the area of the contact surfaces, which was already described by Khaustov et al. [37] for explosion welding. At larger collision angles the temperature was lower due to the lower compression and less heat was transferred to the surfaces. Thus, the potential plastic deformation in the point of collision was reduced at increasing collision angles, which also explains the reduced intensity of the process glare (Figure A1). Moreover, the interaction of both phenomena is conceivable to explain the differences between the cases in Figure 8b,c.

Considering the influence of different flyer thicknesses, for $s = 1.5$ mm a comparable maximum welded area was obtained (ratio: 0.77), whereas for $s = 1.0$ mm the maximum value is smaller (ratio: 0.64). In both cases, the joint was produced in a region close to the lower collision angle boundary,

where for *s* = 2.0 mm the bond formation (ratio: 0.62) has not reached its maximum value. Although some pores and melted interlayers were determined in sections of the weld interfaces of smaller flyer thicknesses, which likely resulted from the interaction with the hot CoP, large inclusions of the CoP could not be detected. Since the impact velocity, the collision angle and thus the air flow in the joining gap were comparable, this could not be attributed to a better ejection condition of the CoP. A possible explanation could be that due to the higher energy input more deformation around the point of collision occurred and, thus, also a larger jet was formed. As shown in Figure 8b (2), the jet can spall in particles. Furthermore, Pabst [38] determined that during the collision welding of aluminum, additional heat was induced into the CoP by an exothermic reaction of chipped particles of the base material. If more base material particles were chipped out by the impact, this in turn might cause more heating. The influence was lower for smaller flyer thicknesses and resulted in less spalled particles, heating and process glare (see Figure A1).

Concerning the governing bond formation mechanisms and considering the SEM images, the largest welded area for *s* = 2 mm was likely produced in the test rig experiments by solid-state welding, whereas excessive melting prevented welding at smaller collision angles. In contrast, the largest welded areas for the smaller flyer thicknesses were formed at least partly by fusion-like bonding. Both processes were strongly influenced by local interaction of the surfaces with the CoP.

These findings are transferable to the MPW interface (Series 2.1, 2.2, 2.3). A certain angle had to be exceeded to allow the ejection of the CoP and to initiate bond formation. The weld interface started with a partially melted but bonded interface where parts of the CoP were entrapped which is also supported by the findings in [6,39]. The further transient rolling movement of the flyer facilitated the ejection of the CoP, whereas the formation of the jet by plastic deformation occurred more intensively at first and then weakened again. This resulted in a jet behavior similar to that at large collision angles in the test rig (see Figure 9) (6). These sections were not clearly distinguishable in the SEM images. Considering the locations of weld defects by melting or CoP-entrapment, it is assumed that in the case of 1.5 mm, most of the weld interface was a result of fusion-like bonding, whereas the latter sections were bonded by solid-state welding due to the occurrence of the jet at the end. In the case of 2 mm flyer thickness, solid-state bonding dominated the weld interface due to the absence of such weld failures in the interface. On the other hand, a jet at the end of the weld interface was not clearly visible (Figure 9) (4).

Despite these different bond mechanisms and resulting weld interfaces, no significant dependence of the joint strength on bond mechanisms was identified (see Figure 11). The differences in the determined tensile forces mostly depended on the weld interface width and the question of whether a complete ring-elliptical weld interface was formed or not, which directly influences the size of the welded area (compare with Figure 11).

*4.3. Prediction and Control of the Weld Interface's Formation and Properties*

The results in Appendix A show that the process glare detected during the MPW experiments was brighter for an increased flyer thickness (Series 2.1, 2.2, 2.3). This is in good agreement with the experiments performed in the test rig where the kinetic collision conditions were kept constant. It is likely that the increased kinetic flyer energy led to an increased formation of CoP (or jet) as also described by Eichhorn [33]. Figure A2 shows the tendency of an increased flash intensity for smaller acceleration distances, which points to the conclusion that the collision angles were smaller and, thus, the compression and light emission of the CoP (or jet) were intensified. This finding can also be related to higher temperatures, as reported by Bellmann et al. [35].

The correlation of process glare parameters with the welding results revealed that the longest weld seams and the highest bond strengths were achieved in experiments with the longest and brightest impact flashes. This was the case for the highest flyer thickness and energy input, respectively, as well as for the smallest acceleration distance. Although the impact flash is only a *necessary* but not a sufficient welding criterion, this observation underlines the importance of a sufficient surface activation prior to

surface contact, which comes along with a bright impact flash. At this point, it should be noted once more that only the impact velocity was adjusted directly during the MPW experiments, but not the collision angle. Thus, the reason for the increased weld length and strength obtained by using thicker flyers and smaller acceleration distances might also be attributed to the differences in the collision kinetics, especially the reduced collision angle. Starting with a small collision angle and assuming similar progression rates, a larger portion of the flyer collides under weldable conditions with the target compared to a high initial collision angle. This led to longer weld seams and brighter impact flashes. Hence, the correlation between the impact flash and the weld strength reported here is valid and appears to be a powerful tool for process development and quality assurance during MPW processes.

Considering the largest ratios of welded to the overlapping area at the test rig (Figure 7), it can be derived that there is a collision angle region leading to optimum collision welding conditions. This region increases and is shifted to larger angles when the kinetic energy input is increased. This kinetic energy input can be varied via the impact velocity [24] or the flyer thickness. The resulting adjustment of the progression rate of the collision angle allows controlling location and length of the weld interface during MPW.

## 5. Conclusions and Outlook

The collision kinetics influence both the CoP formation and its temperature (measured by analyzing the process glare in the companion paper [10]). It, therefore, determines the governing bond mechanism and thus, the reachable amount of welded area. The latter is, however, also influenced by other parameters like (initial) collision angle, its progression and the rolling movement.

Depending on the process conditions, the CoP can be useful for, or harmful to, the bond formation, it furthermore determines the predominant bonding mechanism.

The results of the test rig experiments confirm that the width of the weld interface can be increased by a smaller gradient of the collision angle, when the weldable area of the welding window is reached. Therefore, it could be useful to prepare the flyer geometry to influence its rolling movement on top of the target during MPW and, thus, to improve the weld interface formation. Together with the acquired knowledge about the different ways to increase the energy input, it is possible to adjust the size and location of the weld interface by setting the process parameters. In addition, monitoring of the process glare potentially enables quality assurance of high-strength joints.

Considering that the amount of the CoP increases continuously with the length of the colliding joining partners, the question arises in which areas it is still possible to consider a stationary welding process despite constant kinetic conditions. Therefore, it is of interest for further investigations if and how the local bond properties change and if the welding window can be extended by considering, as an additional factor, the quality of the weld interface.

**Author Contributions:** Conceptualization, B.N., E.S., J.L.-A., J.B. and M.B.; methodology, data analysis, B.N., E.S., J.L.-A. and J.B.; design of experiments, investigation, validation, B.N. (test rig experiments, ultrasonic investigation, long time exposures, SEM analysis), E.S. (MPW experiments, tensile shear tests, optical microscope and SEM analysis), J.L.-A. (PDV-measurement, flash measurement), J.B. (flash measurement) and M.B. (characterization of material's properties); writing—original draft preparation, visualization, B.N., E.S. (Sections 2.3, 2.5, 3.2, 4 and 5), J.L.-A. (Sections 2.3, 2.4.2, 2.4.3, 4 and 5) and J.B. (Sections 2.4.4, 4 and 5); writing—review and editing, S.B., A.E.T., C.L., M.F.-X.W. and P.G.; resources, supervision, project administration, funding acquisition, S.B., A.E.T., E.B., M.F.-X.W., and P.G. All authors have read and agreed to the published version of the manuscript.

**Funding:** This research was funded by the Deutsche Forschungsgemeinschaft (DFG, German Research Foundation), grant number BE 1875/30-3, TE 508/39-3, GR 1818/49-3, WA 2602/5-3, BO 1980/23-1 and is based on the results of the working group "high-speed joining" of the priority program 1640 ("joining by plastic deformation"). The working group consists of the subprojects A1, A5, A8 and A9. We also acknowledge support by the German Research Foundation and the Open Access Publishing Fund of the Technical University of Darmstadt for providing this publication as open access.

**Acknowledgments:** The experiments at the model test rig were conducted at the Bachelor Thesis of Yabing Wang under the supervision by Benedikt Niessen. Moreover, we would like to thank Walter Tutsch of PCO AG for the support in setting up the image intensifier camera. The authors also greatly appreciate the help of Stephan

Ditscher of Baumüller who supported the programming of the system control of the test rig. We thank Ammar Ahsan for his efforts in proofreading the manuscript.

**Conflicts of Interest:** The authors declare no conflict of interest. The funders had no role in the design of the study; in the collection, analyses, or interpretation of data; in the writing of the manuscript, or in the decision to publish the results.

## Appendix A  Analysis of the Process Glare

The qualitative examination of the process glare during test rig experiments is represented in Figure A1 for flyer thicknesses of 1 mm and 1.5 mm at different collision angles. It should be noted that the majority of process glare at the images was visible after leaving the joining gap and no temporal resolution is shown. Nevertheless, it can be seen that the intensity increases with the thickness of the flyer and the glare's color changes from red-orange to white-blue. Furthermore, the collision angle had a significant influence on the intensity, color and amount of the process glare. For angles lower than the lower boundary angle, the glare in the gap was not visible. Therefore, the glare was visible at the sides of the colliding joining partners with only weak intensity. With increasing collision angle its intensity and shape also increased up to a maximum. At larger collision angles the shape remained at the same level, but the intensity decreased, especially when the upper boundary angle was exceeded.

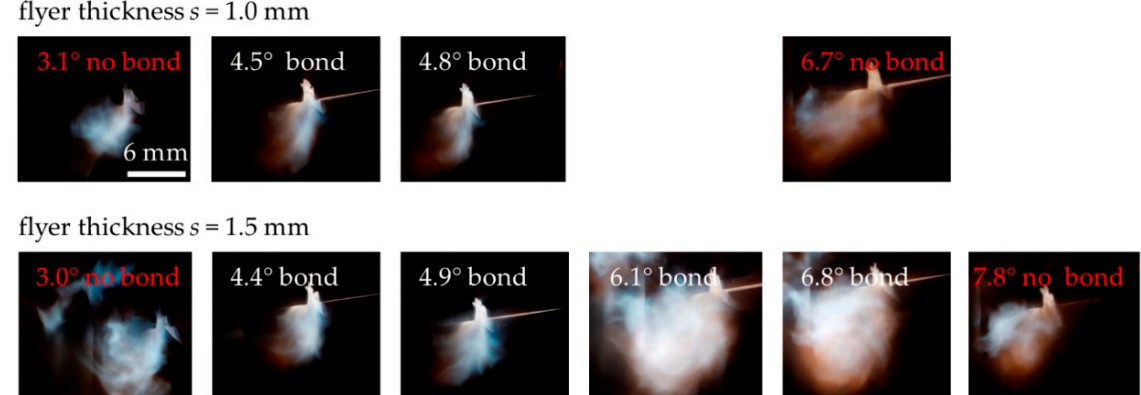

**Figure A1.** Comparison of the process glare for 1 mm and 1.5 mm flyer thickness at different collision angles, recorded by longtime exposure.

As mentioned in Section 2.4.4, the weld formation was accompanied by a bright flash when impact welding is performed in ambient atmosphere. The average values of the impact flash duration and its maximum intensities were evaluated for each parameter set (see Figure A2). The flash duration as well as the maximum intensity increased with higher flyer thickness. The influence of the acceleration distance on the light emission was not that clear due to the data spread. There is a slight tendency showing a decrease in the flash duration and intensity for increased acceleration distances.

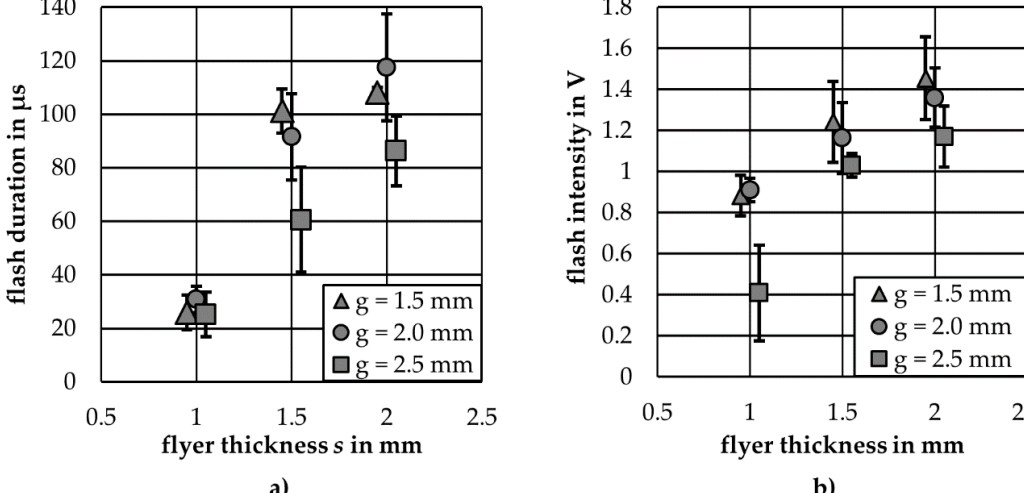

**Figure A2.** (**a**) Increasing flash durations and (**b**) increasing intensities for increasing flyer thicknesses (1.0 mm, 1.5 mm and 2.0 mm) at different acceleration distances *g*. Each point represents four or five experiments with the corresponding standard deviation.

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
