# Peer review of "Interface Formation during Collision Welding of Aluminum"

_metals, doi:10.3390/met10091202_

Round 1

Reviewer 1 Report

Review report

on the manuscript entitled “Interface formation during collision welding of aluminum” by authors B. Niessen, E. Schumacher, J. Lueg-Althoff, J. Bellmann, M. Boehme, S. Böhm, A.E. Tekkaya, E. Beyer, C. Leyens, M.F.-X. Wagner and P. Groche (Manuscript code: metals-905340)

In this study, technical aspects of relatively new magnetic pulse welding (MPW) were considered. In my opinion, this is an excellent work, which represents an interest for welding society and worthy for publication. I have only minor recommendation to the authors (please add the missed “1” label to Fig. 8d).

In their future works, I recommend the authors to pay more attention to examination of the microstructural aspects of this joining technique. Considering a very specific thermos-mechanical conditions arising in the weld zone, the resulting microstructure may represent a considerable academic interest.    

Author Response

Dear reviewer,
thanks for your assessment and helpful comments. The missing label in Figure 8d) was supplemented.

We agree that the initial state and the development of the microstructure have an enormous influence on bond formation. Please find a first approach of the determination of the thermal conditions in our companion paper (Bellmann, J.; Lueg-Althoff, J.; Niessen, B.; Böhme, M.; Schumacher, E.; Beyer, E.; Leyens, C.; Tekkaya, A.E.; Groche, P.; Wagner, M.F.-X.; et al. Particle Ejection by Jetting and Related Effects in Impact Welding Processes. Metals 2020, 10, 1108, doi:10.3390/met10081108).

Kindest regards

Reviewer 2 Report

This manuscript dealt with the interface formation during magnetic pulse welding. First of all, it is quite misleading to use the term Collision Welding - in North America, it means welding for repair collision! I would like to suggest the authors to consider to change the title of their manuscript to be "Interface formation by collision during magnetic pulse welding". As such, the authors should carefully revisit and change accordingly their use of this term through out the manuscript.

Author Response

Dear reviewer,

thanks for your helpful comments. Meanwhile, the manuscript was linguistically corrected. We agree that an accurate definition of processes is important in the scientific and industrial community. The term collision welding is used because it describes not only welding by the impact of one joining partner at another but also the collision of two moving joining partners as it occurs e.g. in the test-rig. Therefore, we prefer to use this term which is also commonly used by a research group of the Ohio State University:
Vivek, A., Hansen, S. R., Liu, B. C., & Daehn, G. S. (2013). Vaporizing foil actuator: A tool for collision welding. Journal of Materials Processing Technology, 213(12), 2304-2311.
Vivek, A., Liu, B. C., Hansen, S. R., & Daehn, G. S. (2014). Accessing collision welding process window for titanium/copper welds with vaporizing foil actuators and grooved targets. Journal of Materials Processing Technology, 214(8), 1583-1589.

Kindest regards

Reviewer 3 Report

Please see the attached PDF file including my comments.

Author Response

Dear reviewer,

thanks for your helpful and detailed comments. Please consider our comments in the manuscript and the following answers to your comments:

1. We agree that several other parameters like e.g. material (combination) of the joining partners influence the possible welding mechanisms and the relevance of the horizontal collision point velocity. This parameter is of great importance for the formation of the jet and must be lower than the speed of sound inside of the colliding material. Furthermore, together with the collision angle, it is relevant for the compression and the re-entrapment of the cloud of particles (CoP) in the joining gap. But the collision point velocity is related trigonometrically with collision angle and impact velocity, which determine the force or the pressure at the collision point. This relationship is also valid for MPW. Futhermore, we agree that we do not know the exact values of the parameters during the highly transient welding process. However, we can deduce their development from former numerical and experimental research and we described the parameter progression in the first paragraph of section 4.1. At the initial contact, the collision angle is close to 0°, therefore, the collision point velocity is theoretically close to infinite. During the further progression, the collision angle increases, and the impact velocity decreases in general. Each experiment with the test rig with a certain collision angle and impact velocity represents the process condition at a specific moment during the MPW process.
Your conclusion is plausible for the experiments with 1 mm and 1.5 mm flyer thickness: the highest impact energy and force at the point of collision (PoC) due to the smallest acceleration gap lead to the largest summed width of the weld interfaces and highest achieved tensile force. But for 2 mm flyer thickness, the values of summed width of the weld interfaces and tensile force are similar for 1.5 mm and 2 mm acceleration gap. However, the position of the weld interface is different due to the different progression of the process parameters through the weldable region of the welding window. The lower deviation of the tensile force for the acceleration gap of 2 mm in comparison to 1.5 mm can be related to less re-entrapment of the CoP due to the comparable larger collision angle which leads to less local defects in the weld interface.

2. We also discussed this point during the writing process and understand your remark. We decided for these representations because in Fig. 10 we want to highlight on the one hand the increasing summed width of the weld interface with increasing flyer thickness. On the other hand, the decrease of the gap width with increasing acceleration gap is visible. In Fig. 11 we chose the presented order due to the flyer thickness dependent theoretical bearable tensile forces which were calculated by the quotient of tensile strength and cross-sectional area of the flyer sheet.

3. The used aluminum alloy EN AW-1050A HX4 is a pure aluminum alloy in a half-hard condition. The strain hardening condition of such an aluminum alloy is adjusted by cold rolling. A substantial amount of work hardening is not expected in the used material. A change of grain due to thermal effects is located very narrow to the collided surfaces and can be neglected for the mechanical failure (Compare: Khaustov, S.V.; Kuz’min, S.V.; Lysak, V.I.; Pai, V.V. Thermal processes in explosive welding. Combust. Explos. Shock Waves 2014, 50, 732–738, doi:10.1134/S0010508214060161). Furthermore, the following sentence was added to the manuscript: “In this case the fracture occurred in the neck region of the flyer close to the welded area, where the cross-sectional area was reduced due to plastic deformation during the welding process.“ Additionally, a fracture image of the failure in the base material was complemented in Fig. 11.

4. The typing error in line 424 was corrected. No error could be found in line 61.

5. The abbreviation “PDV” is introduced in the capture of Fig. 3.

6. The structure of chapter 2 is rearranged to improve accessibility. Furthermore, Table 1 is added with a summary of the parameters that were changed and kept constant for each series of experiments. The enumeration of Table 1 is complemented in the text to simplify allocation.

7. The value of the impact velocity is added to Fig. 7 and Fig. 8.

8. “Summed width” and “gap width” are added to Fig. 9.

9. The arrow of impact velocity was removed from Fig. 12.

10. Please consider our answer to comment 1. We agree that the force or pressure at the CoP increases with decreasing acceleration gap and constant impact velocity due to the smaller collision angle. But due to the continuous progression of the process parameter the pressure distribution and, in addition, the flow profile in the joining gap during the collision process is unknown and must be considered in further research. This will allow to determine the true conditions at the collision point and their influence on the governing bond mechanisms.

Kindest regards

Round 2

Reviewer 2 Report

The authors addressed my comments.